# Concurrent Detection of a Papillomatous Lesion and Sequence Reads Corresponding to a Member of the Family *Adintoviridae* in a Bell’s Hinge-Back Tortoise (*Kinixys belliana*)

**DOI:** 10.3390/ani14020247

**Published:** 2024-01-12

**Authors:** Johannes Hetterich, Monica Mirolo, Franziska Kaiser, Martin Ludlow, Wencke Reineking, Isabel Zdora, Marion Hewicker-Trautwein, Albert D. M. E. Osterhaus, Michael Pees

**Affiliations:** 1Department of Small Mammal, Reptile and Avian Medicine and Surgery, University of Veterinary Medicine Hannover Foundation, Bünteweg 9, 30559 Hannover, Germany; michael.pees@tiho-hannover.de; 2Research Center for Emerging Infections and Zoonoses (RIZ), University of Veterinary Medicine Hannover Foundation, Bünteweg 17, 30559 Hannover, Germany; monica.mirolo@tiho-hannover.de (M.M.); martin.ludlow@tiho-hannover.de (M.L.); albert.osterhaus@tiho-hannover.de (A.D.M.E.O.); 3Department of Pathology, University of Veterinary Medicine Hannover Foundation, Bünteweg 17, 30559 Hannover, Germanyisabel.zdora@tiho-hannover.de (I.Z.); marion.hewicker-trautwein.ir@tiho-hannover.de (M.H.-T.)

**Keywords:** reptile medicine, emerging diseases, neoplasia, next-generation-sequencing, metagenomic examination, surgery, adintovirus

## Abstract

**Simple Summary:**

This article describes the diagnostic evaluation and therapy of a tortoise with a bulging oral lesion of unknown origin. Initially, the animal owner indicated that a similar oral mass at the same location had been surgically removed four years ago. At that time, the abnormal oral structure had not been evaluated further. This time, the mass was removed surgically and further examined by both veterinary pathologists (veterinarians specialised in the examination of animal diseases and body structures) and virologists. No causal agents, which have been described before to trigger comparable lesions in tortoises, were found. Parts of the removed lesion were examined in further virological studies to scan the tissue material for potential new infectious agents, which might be connected to the abnormal oral lesion. Indeed, the authors were able to detect virus material within the tissue mass belonging to a comparably new virus family. The exact influence of these agents on the origin of the lesion, though, remains unclear.

**Abstract:**

An adult male Bell’s hinge-back tortoise (*Kinixys belliana*) was admitted to a veterinary clinic due to a swelling in the oral cavity. Physical examination revealed an approximately 2.5 × 1.5 cm sized, irregularly shaped tissue mass with villiform projections extending from its surface located in the oropharyngeal cavity. An initial biopsy was performed, and the lesion was diagnosed as squamous papilloma. Swabs taken for virological examination tested negative with specific PCRs for papillomavirus and herpesvirus. Further analysis of the oropharyngeal mass via metagenomic sequencing revealed sequence reads corresponding to a member of the family *Adintoviridae*. The tissue mass was removed one week after the initial examination. The oral cavity remained unsuspicious in follow-up examinations performed after one, five and twenty weeks. However, a regrowth of the tissue was determined 23 months after the initial presentation. The resampled biopsy tested negative for sequence reads of *Adintoviridae*. Conclusively, this report presents the diagnostic testing and therapy of an oral cavity lesion of unknown origin. The significance of concurrent metagenomic determination of adintovirus sequence reads within the tissue lesion is discussed.

## 1. Introduction

In chelonians, aetiologies for tissue proliferations including neoplastic diseases are diverse [1,2,3]. Oropharyngeal swelling can generally be associated with papilloma-like lesions, which have been described several times for various reptile species [4]. Papillomaviruses have been found in skin lesions [5], also causing cutaneous papillomas in sea turtles [6]. However, the presence of papilloma-like viral sequences has not been linked to clinical symptoms in all cases. The field of reptilian virology has progressed significantly over the last two decades [4,7,8,9,10]. Standardisation of sampling methods and increased use of molecular diagnostics—particularly the use of next-generation sequencing (NGS)—has led to a marked increase in the number of described viruses in reptile medicine [4]. Also, the understanding of the viral impact in specific diseases has been enhanced in recent years. However, the interconnections between specific viruses and many extrinsic and intrinsic factors, such as the reptilian immune system, environmental conditions and coexisting infectious diseases, which contribute to a clinical illness remain unclear for a large number of viral diseases [11].

This is the first report of a Bell’s hinge-back tortoise with a distinct clinical symptom alongside concurrent metagenomic determination of sequence reads corresponding to a member of the family *Adintoviridae*.

## 2. Material and Methods

### 2.1. Animals

Initially, an adult male Bell’s hinge-back tortoise (*Kinixys belliana*), weighing 911 g, being approximately 35 years old and originating from Benin, West Africa, was presented (index case patient). The tortoise had a one-week history of reduced forage intake and diarrhoea. Anamnestically, the care worker indicated a former tissue enlargement in just the same oropharyngeal location that had been removed by a veterinarian four years before. At that time, the tissue sample did not undergo any further diagnostic examination. The tortoise had been kept in an enclosure (six square metres) with regular access to the suburban garden during the summer season for the last 25 years. It had been fed a variety of salads and vegetables and had received regular mineral supplementation as well as species-appropriate UVB light.

After the initial identification of adintovirus sequences in the index case, a population of Bell’s hinge-back tortoises under human care were sampled to evaluate the presence of similar (adinto)virus sequences in conspecifics. The population included nine tortoises: two individuals from a German zoo and seven animals (five zoo animals and two privately kept tortoises) from Zurich, Switzerland. All animals were raised under human care and evaluated as adult tortoises in clinically healthy condition. No oral lesions or other abnormal findings were reported for any of the nine sampled tortoises.

### 2.2. Diagnostics

Initial diagnostic methods of the index case patient included a tissue biopsy for histopathological examination, obtaining oral and cloacal swabs for virological examination as well as oral swabs for microbiological and mycological examination, parasitological faecal examination and blood chemistry examination. Further diagnostic evaluation involved next-generation-sequencing (NGS) examinations of whole blood, cloacal and skin swabs and faeces to determine the presence of viral sequences.

Sampling of the nine conspecifics comprised cloacal and skin swabs, faeces and whole blood. The samples of these nine animals were only examined for next-generation-sequencing analysis.

For histopathological examination, the tissue sample was fixed in 4% neutral buffered formalin for at least 24 h. The formalin-fixed tissue was dehydrated and routinely embedded in paraffin wax, sectioned at 4 µm and stained with haematoxylin and eosin (HE). Slides were evaluated with a standard binocular light microscope (Carl Zeiss 670; Carl Zeiss, Jena, Germany; field of view area, 40× magnification: 0.16 mm^2^). For image preparation, the specimen was digitalised using an Olympus VS200 slide scanner (Olympus Deutschland GmbH, Hamburg, Germany), and representative images were exported with the respective OlyVIA software (version 3.4.1., Olympus Deutschland GmbH, Hamburg, Germany).

For bacteriological and mycological examination, swabs were inoculated on different culture media (Columbia agar with sheep blood, Gassner agar, Staphylococcus/Streptococcus selective agar, neomycin agar, chocolate agar, Schaedler agar, Kim-mig agar) and incubated under aerobic, anaerobic or microaerophilic atmosphere at 37 °C for up to 48 h. The Kim-mig agar was incubated for 48 h at 30 °C. The swab was further placed in nutrient broth, which was also stored at 37 °C overnight. The following day, the nutrient broth was spread onto selected culture media (Columbia agar with sheep blood, Gassner agar, Staphylococcus/Streptococcus selective agar), which were incubated at 37 °C for an additional 24 h. The different colonies grown on the culture media were differentiated by mass spectrometry (MALDI-TOF, Bruker, ORT).

Blood chemistry examination was performed in a standardised manner in an inhouse laboratory (Cobas C 311, La Roche Ltd., Basel, Switzerland) 15 min after venipuncture (dorsal coccygeal vein). Haematological parameters were examined microscopically at 1000× magnification after standardised blood smear preparation with 20 uL whole blood.

For NGS examination, a section of the tumour tissue sample (20 mg) was homogenised in 500 µL PBS and centrifuged for 5 min at 12,000 relative centrifugal force (RCF). The supernatant was sieved using a 0.45 µm spin filter (Merck Millipore, Darmstadt, Germany) for 5 min at 12,000 RCF. RNA was extracted using TRIzol Reagent (Thermo Fisher Scientific, Dreieich, Germany) in accordance with the manufacturer’s instructions and temporarily stored in a −80 °C freezer. Conversion of RNA to cDNA was performed using Superscript IV (Thermo Fischer Scientific, Dreieich, Germany) and nonribosomal hexamers as per protocol guidelines. Double-stranded DNA (dsDNA) was generated via the Klenow fragment (NEB), followed by random PCR amplification using a sequence-independent, single-primer amplification protocol [12]. PCR products were purified a using Monarch PCR/DNA Cleanup kit (NEB), and a DNA library was prepared with a Nextera XT DNA Library Preparation Kit (Illumina Inc., San Diego, CA, USA) prior to sequencing on an Illumina NextSeq sequencing platform with a NextSeq 500/550 High Output kit v2.5 for 150 cycles (paired-end reads, 75 base pairs). Raw sequencing data (FASTQ) were first analysed using the ID-Seq bioinformatics pipeline (https://czid.org/) [13,14]. Additional bioinformatic analyses were performed in CLC Genomics Workbench (v12) (QUIAGEN).

Further virological investigation was attempted by extracting RNA from blood (whole blood), cloacal swabs, skin swabs and faeces. Primers (Adinto_Integrase_frw 5′-TGCTCATGTCTGAGACACAGATATC-3′ and Adinto_Integrase_rvs 5′-CTACAGAGCTGGCATTGCTG-3′) were designed based on the recovered partial adintovirus sequence to confirm the presence of an adintovirus in the RNA which had been used for NGS analysis and to complete the sequence of the integrase gene. PCR amplification was performed using a Qiagen One Step RT-PCR kit (Qiagen, Germantown, MD, USA) and an annealing temperature of 49 °C. Additional screening of blood (whole blood), cloacal and skin swabs and faeces for the presence of adintovirus was performed according to the same primers and protocol.

## 3. Results

### 3.1. Initial Clinical Condition and Therapy

Clinical examination of the index case patient revealed an approximately 2.5 × 1.5 cm sized, irregularly shaped tissue mass with villiform projections extending from its surface in the left oropharyngeal cavity (Figure 1a). In addition, the animal showed liquid defaecation throughout the general examination. A subsequently performed faecal examination revealed flagellate overgrowth.

Initially, the index case tortoise received treatment for flagellate overgrowth (metronidazole 40 mg/kg PO; repeated after 14 days; Eradia, Virbac Tierarzneimittel GmbH, Bad Oldesloe, Germany). Furthermore, a supportive therapy was started with a liver-supporting product (50 mg/kg SID PO; Legaphyton^®^; Vetoquinol S.A., Lure, France) and probiotics (0.5 mL solution PO; BeneBac^®^, Dechra Veterinary Products Deutschland GmbH, Aulendorf, Germany). One week later, the tortoise was presented for a follow-up and a subsequent surgical extirpation of the oropharyngeal tissue enlargement. The patient’s overall condition had improved. It showed regular voluntary food consumption, and the faecal quality was reported to have a more solid consistency. The animal received analgetic pain therapy with meloxicam (0.3 mg/kg SC; Metacam, Boehringer Ingelheim Vetmedica GmbH, Ingelheim, Germany) and lidocaine (1 mg/kg topical; Lidor^®^ 2%, VetViva Richter GmbH, Wels, Austria) and was then sedated with alfaxalone (5 mg/kg IV; Alfaxan^®^, Jurox Pty Limited, Rutherford, Australia). The tortoise maintained spontaneous breathing during the 15 min procedure and was therefore not intubated. The oral tissue mass was removed without any macroscopic leftover tissue using forceps and a sharp spoon (Figure 1b). Only minor bleeding (approximately 0.3–0.5 mL) occurred inside the oral cavity, and the animal regained full activity within 45 min after the surgical procedure.

### 3.2. Diagnostic Results

Blood chemistry examination revealed high levels of liver-linked enzymes GLDH (75.4 U/L), AST (521 U/L) and ALT (35 U/L). Haematological parameters were considered unremarkable.

Histopathological analysis revealed fingerlike projections of keratinising squamous epithelium with marked hyperplasia and ortho- to parakeratotic hyperkeratosis (Figure 2A,B). Regular maturation of keratinocytes was retained. Koilocytes were not present. Subepithelially, low amounts of fibrovascular stromata were present. There was mild anisocytosis and karyosis. In ten high-power fields (total field of view: 1.6 mm^2^), one mitotic figure was present in the suprabasal epithelium. Furthermore, there was marked, multifocal, superficial, intraepithelial, supportive inflammation with multifocal small colonies of coccoid bacteria. Based on the histopathological findings, a squamous papilloma was diagnosed.

Microbiological assessment revealed a diverse bacterial flora including *Morganella morganii*, *Pseudomonas aeruginosa* and *Streptococcus* sp. No fungi were cultured. Microscopic faecal examination revealed a high infestation of flagellates (flagellate overgrowth). No further parasitological pathogens were found. Virological examination of dry swabs for papillomavirus and herpesvirus using family-specific PCR methods [15,16] were without positive confirmations.

Analysis of fastq raw reads using the CZ ID metagenomics pipeline [13,14] showed the presence of 23 non-genus-specific reads showing homology to Terrapene box turtle adintovirus (GenBank accession no. BK010890.1), a member of the *Adintoviridae* family. Reference assembly using Genomics Workbench (v12) by read mapping to the Terrapene box turtle adintovirus sequence (GenBank accession no. BK010890.1) enabled the identification of an additional 913 adintovirus reads. The complete integrase sequence was obtained by Sanger sequencing of RT-PCR amplicons generated using primers based on adintovirus sequences found in NGS analysis. A nucleotide BLAST (BLASTN) search of already known adintovirus species indicated that the adintovirus present in the clinical tortoise sample was closely related to the Terrapene box turtle adintovirus, with 95.16% nucleotide identity, followed by multiple uncharacterised mRNAs from related turtle species. To better distinguish the newly identified adintovirus from closely related viruses, we used the nomenclature ‘Bell’s hinge back tortoise (*Kinixys belliana*) adintovirus’. In addition to the integrase gene, analysis of NGS data enabled the recovery of a contig of 242 base pairs (bp) of the penton gene, 490 bp of the genome packaging ATPase and 457 bp from the DNA polymerase gene, with 88.84%, 94.94% and 93.44% nucleotide identity, respectively, to the homologous Terrapene box turtle adintovirus genes. An RT-PCR using specific integrase primers resulted in a detectable amplicon verified subsequently by Sanger sequencing in one additional sample from the index case patient (skin swab). Additionally, two out of nine sampled tortoises (once each for a cloacal swab and whole blood) tested positive for adintovirus RNA.

However, the complete adintovirus genome could not be recovered by Sanger sequencing and reference assembly. Furthermore, no oncogene transcripts were identified by NGS or Sanger sequencing.

### 3.3. Follow-up History

The index case tortoise was presented for three additional follow-ups at one, five and twenty weeks after removal of the tissue enlargement. The oral cavity remained unremarkable, and no regrowth of tissue in the affected oral region could be identified. The clinical condition continuously improved after the surgical procedure. The tortoise returned to its usual food consumption and activity within five days after surgery. The overall condition was evaluated as good at follow-ups five and twenty weeks after initial presentation. However, a regrowth of the tissue was determined 23 months after the initial presentation. At that time, the tortoise showed regular activity and unsuspicious defecation and food intake. The tissue lesion again was removed surgically according to the procedure described for the initial surgery. Macroscopically, the oral lesion strongly resembled the initial tissue mass. However, the lesion had a considerably smaller (1.0 × 0.5 cm) size and flatter structure. As before, the tissue mass was removed without macroscopic leftovers. The recurring papillomatous tissue was resampled and assayed by RT-PCR, but no adintovirus RNA was detected. No histological examination was conducted. The animal remained in a good clinical condition in subsequent follow-up examinations two and twelve weeks after the second surgical procedure. No further regrowth was determined.

All liver-linked parameters remained markedly elevated in each of the follow-up blood examinations.

## 4. Discussion

The oropharyngeal tissue mass was the primary clinical symptom observed in the tortoise and needs to be discussed as a possible recurrent finding. Based on information given by the patient’s owner, a tissue enlargement in exactly the same location had been surgically removed four years previously in the absence of any further postsurgical examination. According to the owner, the shape, size and colouration of the tissue lesion were comparable to the macroscopical appearance of the initial tissue enlargement. Aetiologies for tissue proliferations, particularly of mucous membranes, are diverse in reptile medicine, including a broad variety of neoplastic diseases [1,2,3]. In chelonians, a plethora of neoplasia reports exist based on case reports [11]. Virus infections as causative agents need to be considered, especially since reptilian virology has progressed significantly over the last two decades by standardisation of sampling methods and increased use of molecular diagnostics [7,8,9,10]. The oropharyngeal swelling’s villiform macroscopical appearance could be associated with papilloma-like lesions. These have been described for various reptile species [4]. Papillomaviruses have been found in skin lesions [5], also causing apparent cutaneous papillomas in loggerhead sea turtles and green sea turtles [6]. However, the presence of papilloma-like viral sequences has not been associated with clinical symptoms in all cases. In tortoises, papillomavirus-like particles have been detected in different species, causing papular skin lesions in one case [17]. Nevertheless, papillomaviruses have been described as being highly host- and tissue-specific [4]. The clinical status of papilloma-caused lesions (progression, resolution) has also been found to be highly variable [17]. In this case, no papillomavirus was detected, either by PCR or by next-generation sequencing (NGS). Also, reptilian herpesvirus infections need to be considered for irregular oral cavity findings in tortoises. Several herpesviruses have been described for numerous members of the family *Testudinidae* [4]. If clinical signs of an infection are present, rhinitis and conjunctivitis are oftentimes associated with indicative stomatitis and glossitis, leading to diphtheroid-necrotising alterations in advanced stages of the disease. Diphtheroid membranes covering oral cavity tissue can be seen regularly, as brownish villiform structures initially were covering the oral cavity at the patient’s first presentation. However, the lack of associated lesions (rhinitis, conjunctivitis, glossitis) and the more rigid, bulging appearance of the tissue lesion did not support herpesvirus infection as a differential diagnosis.

Further analysis of a tissue biopsy of the oropharyngeal lesion using NGS resulted in the identification of sequence reads displaying homology to a member of the *Adintoviridae* family, which we termed ‘Bell’s hinge back tortoise (*Kinixys belliana*) adintovirus’. The name ‘adintovirus’ refers to a Polinton, which harbours retrovirus-like integrase proteins, and an adenovirus-like DNA polymerase [18]. The *Adintoviridae* family was first recognised in 2021 and is composed of two lineages, Alpha and Beta. A BLASTN search of the integrase gene showed that the virus is closely related to an adintovirus recovered from a box turtle, called ‘Terrapene box-turtle Adintovirus’. Using data mining, a recent study found that most adintovirus genomes bear open reading frames (ORFs) with predicted structural similarity to capsid hexon and penton proteins of multiple known viruses, including adenoviruses [18]. Therefore, questions arise as to the ability of adintoviruses to produce progeny virions or rather if the adintovirus genome is maintained in the form of an integrated provirus. Although the complete genome of an adintovirus could not be recovered in this study, reference assembly of recovered sequencing reads to the Terrapene box turtle adintovirus enabled the identification of transcripts of the penton and genome packaging ATPase genes. A BLASTN search using the integrase gene of the adintovirus identified in this study showed that the most similar virus was an adintovirus from a related tortoise species, the Terrapene box turtle, followed in the same query unexpectedly by multiple uncharacterised mRNAs from related tortoises. Overall, these observations strongly suggest that adintoviruses are integrated into the respective host genome and have subsequently coevolved over time.

Another key feature of the adintoviruses genome is the presence of ‘oncoid genes’, which are similar to the well-characterised retinoblastoma-interacting protein and antiapoptotic proteins of small DNA tumour viruses, such as polyomaviruses, papillomaviruses, parvoviruses and adenoviruses [18]. However, adintoviruses have so far not been associated with diseases or tumours. On the contrary, transcripts from both adintovirus lineages were identified in the gill tissue of nondiseased freshwater fish by metatranscriptomic analyses [19].

In the present study, we showed that adintovirus transcripts were detectable in the initially sampled papillomatous lesion but also in nonsuspicious tissue (skin swab from the index case patient as well as each cloacal swab and plasma samples of two further clinically healthy tortoises). Related to this, adintovirus as a causative agent of the oral lesion cannot be ruled out. At this time, however, it is not possible to definitely link the presence of adintovirus sequences in the papillomatous lesion to the aetiology of the lesion itself or to the markedly increased liver enzyme elevations and the symptom of diarrhoea. Transmission electron microscopy (TEM) of sections of the papillomatous lesion would be required in order to confirm the presence of adintoviruses virions. In addition, RNA-seq analysis could also be used on appropriate tissue samples from diseased and healthy animals to look for differences in gene expression of adintovirus genes between healthy and diseased animals.

Relating to the coexisting clinical symptoms, various viruses have been described to cause liver disease in reptiles [8,20,21]. In this case, further sampling such as liver biopsy or endoscopic coelomic imaging might have been highly beneficial. However, the animal owner rejected any invasive diagnostic procedures. Although the tortoise’s clinical status improved satisfyingly within five days after the initial surgery, the authors critically reflect on the perioperative management of the patient. A more multimodal pain management might have enhanced an even better postsurgical recovery. Furthermore, intubation of the tortoise might have increased intrasurgical options in case of anaesthesia incidents like major bleedings or apnea.

## 5. Conclusions

This report clearly shows the high potential for molecular diagnostics in reptile medicine. The routine use of advanced diagnostic methods such as NGS will facilitate the identification of new reptilian viruses and possible links to known disease syndromes. Prospective standardised study designs should be attempted, especially involving scientifically underrepresented reptile species.

## Figures and Tables

**Figure 1 animals-14-00247-f001:**
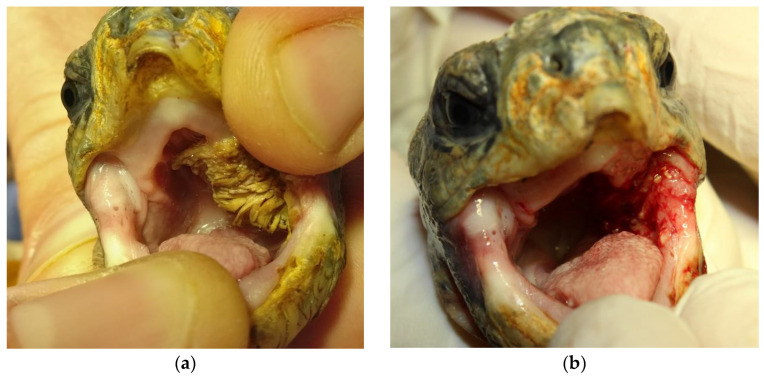
Bell‘s hinge-back tortoise, oral cavity. (**a**) Upper-left side, showing an approximately 2.5 × 1.5 cm sized oropharyngeal villiform tissue enlargement on the left oropharynx. (**b**) Upper-right side, left oropharynx is now visible after complete removal of the villiform tissue enlargement.

**Figure 2 animals-14-00247-f002:**
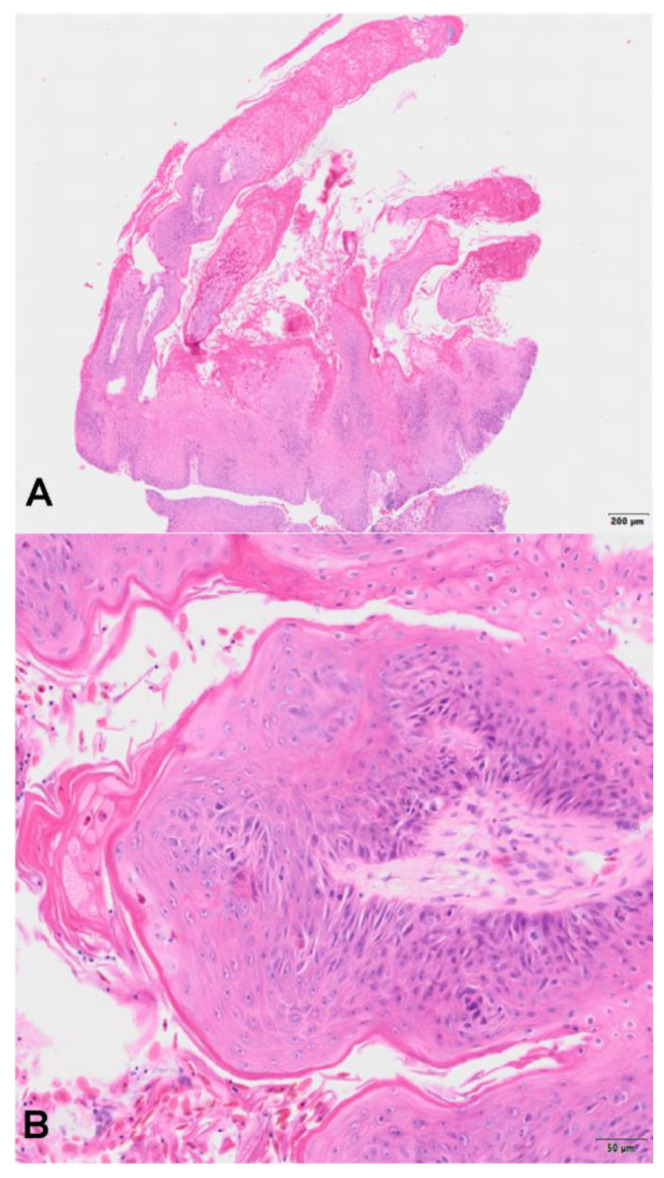
Bell’s hinge-back tortoise (*Kinixys belliana*), oropharyngeal mass. (**A**) Squamous papilloma with thickened, exophytic, hyperplastic and hyperkeratotic epithelium arranged in fingerlike projections. Haematoxylin & eosin (H&E). Low magnification. Bar: 200 µm (**B**) High magnification of the squamous papilloma. Exophytic projections are covered by ortho- to parakeratotic hyperkeratosis. The exophytic projections of squamous epithelium are supported by subepithelial fibrovascular tissue. H&E. Bar: 50 µm.

## Data Availability

The data presented in this study are openly available in a repository at DOI:10.17605/OSF.IO/WZJ68 (https://osf.io/wzj68/).

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
