# Peer review of "Concurrent Detection of a Papillomatous Lesion and Sequence Reads Corresponding to a Member of the Family *Adintoviridae* in a Bell’s Hinge-Back Tortoise (*Kinixys belliana*)"

_animals, 2024, doi:10.3390/ani14020247_

Round 1
Reviewer 1 Report
Comments and Suggestions for Authors
This case report finds a novel virus associated with a significant oral lesion in a tortoise species. Overall, the content of the report is worthy of reporting and the authors have worked hard to ensure that the content is relevant by sampling additional tortoises of the same species, which greatly strengthens the findings and sparks more interest!
I struggled with the flow of the manuscript, as it was not always chronologically presented (for instance the follow up from the tortoise was presented before the diagnostic findings were presented from the initial examination). Further, the methods were presented for all diagnostics out of line with the results of those methods, which was also a bit disorienting. Restructuring to follow chronologically would greatly help with reader flow and answer some of my questions.
The discussion would also benefit from a review of herpes causing mucosal lesions, as was performed for papilloma virus, as this is another cause of oral lesions in reptiles (and one that was tested for).
Specific comments
line 49 - metronidazole is misspelled and does not require capitalization.
Line 51 - Legaphyton is a trade name, so should be in the parentheses. I am assuming that this product was initiated and the dose was based upon the silybin?
Line 52 - Benebac is product, and should be in parentheses, replace Benebac in sentence with "probiotics"
Line 56 - consistence = consistency
Line 57 - alfaxalone is misspelled and does not require capitalization.
Additional basic information about the surgical/anesthetic process is warranted.
-How much blood loss (estimated) was present? What was approximate length of procedure.
-Please provide information on what instrument the "sharp curette" was?
-Was analgesia provided? if not, please justify.
-Was the tortoise intubated? If not, please justify
Line 59: "Further tissue samples" sounds like additional biopsies were taken. I suspect that this was not the case and instead the biopsy was divided and placed into formalin and freezer for analyses? Dry swabs taken from the oral cavity or from the biopsy or both?
Line 65: Preliminary and conclusive are pretty opposing terms. That being said, this sentence can read "The overall condition was evaluated as good at the follow up 20 weeks after initial presentation".
Line 72: Mention of cytology being performed but no results listed. The comment comes in the middle of the methodology surrounding histopath, so is a bit out of place.
line 81: coccid should be coccoid?
Line 91: consider "inoculated" instead of "placed"
Line 102: infestation is usually reserved for discussion of parasitic infections. "High-grade infestation of" can be removed from this sentence without changing its meaning.
Line 105: the primers used (gene targets) for these PCRs must be reported and the general techniques. This can be referenced if there is a publication.
Line 111: can remove "any" and the sentence still holds meaning.
Line 151: "off spring breeds" - does this mean "captive bred"??
Line 152: I would recommend directly stating that the 7 additional tortoises lacked oral lesions.
Line 155: How was this confirmed to be a rectal swab? Im suspicious this was actually a cloacal swab?
Line 155: The way this sentence ends, it sounds like a single tortoise was sampled, but the beginning of the paragraph sounds like 9 tortoises were sampled?
Line 166: "clocal" = "cloacal"?
Line 167: Was blood (whole blood) or plasma tested?
Line 168: assuming that resampling occurred at 23 months? Adding in clinical information about follow up for this patient would be nice, given that it is likely that there are several months of follow up after last presentation.
Line 181: increased is misspelled.
Fig 2/3: Please remark on stain (HE likely) and magnification in the legend. Consider joining these two images into one large image with A and B sections.
Line 193 - what was submitted for NGS? Tissue or dry swab or both? If both, the agreement between the 2 samples should be reported
Comments on the Quality of English LanguageThere are some English language concerns that I have tried to help with in some comments below.
Reviewer 2 Report
Comments and Suggestions for Authors
In your initial treatment description you don’t explain why you put the patient on metronidazole. Just ad another sentence that you took a fecal sample and diagnosed flagellan overload, otherwise your reader might think you based your treatment on the clinical symptoms (diarrhea). You only mention the fecal examination later in your text.
Please rethink your anesthesia protocol. Alfaxalon is adequate for initial sedation but has no adequate pain management for surgery. In the information and directions for use alfaxalon is described only as low level analgesic. It has to be used in combination with other analgesic medications for example butorphanol or at least in combination with local anesthesia.
Reviewer 3 Report
Comments and Suggestions for Authors
Dear Editor,
The manuscript “Concurrent detection of a papillomatous lesion and sequence reads corresponding to a member of the family Adintoviridae in a Bell´s hinge back tortoise (Kinixys belliana)” by J. Hetterich with co-authors brings new genomic data about novel virus from the family Adintoviridae isolated from tortoise Kinixys belliana. The authors provided interesting and important data on finding the cause of a disease in a Bell´s hinge back tortoise, which ultimately leads to the discovery of a new virus that affects reptiles. It was possible after analyzing deep sequensing of genomes during metagenomic research. In our laboratory, we also conduct research to find new viruses from amphibians and reptiles out of metagenomic data and imagine that the authors had to do painstaking work to understand the true cause of the disease. Of particular note is the arsenal of methods and approaches of bioinformatics analysis, which made it possible to separate the host genome and the virome, and, ultimately, to identify the virus.
While I support this paper for publication, I have several recommendations designed to help strengthen the study or its presentation. All of these are eminently feasible I suspect.
There are some minor aspects and corrections that should be considered.
84, 89 – italics in Latin names.
Articles in the Animals Journal have the structure “Introduction”, “Materials and Methods”, “Results” and “Discussion”. I ask the authors of the article to adapt it to the standard article structure. Include a description of the case in the “Results” section. Everything related to methods should be included in “Materials and Methods”.
167 – it would be good to sequence the fragments. But if it is not possible to add to this article, then it is at the discretion of the authors. In the future, this information could help in understanding the spread of new emerging diseases.
In general, I think the scope of the manuscript “Concurrent detection of a papillomatous lesion and sequence reads corresponding to a member of the family Adintoviridae in a Bell´s hinge back tortoise (Kinixys belliana)” and its findings fit with the aims of Animals Journal and should be of interest to its readers. The work is well structured, written in a scientific style, competently. The article gives the impression of a completed well-described study and can be accepted after minor corrections.
My conclusion is the acceptance of the manuscript after minor revision.
Best regards,
Round 2
Reviewer 1 Report
Comments and Suggestions for Authors
Thank you for responding to comments to improve manuscript. The formatting changes do allow for increased readability. I have a few additional comments to consider to further improve readability.
1. Consider breaking the sections into headings. For instance, under MM, "Animals" could describe the individual and population of tortoises used. Then "Diagnostics" could be the sub section to describe the detail of the diagnostics performed.
2. The precise sample (or samples) taken, tested and with specific results is not always clear. I would recommend closely evaluating the
3. Consider calling the initial tortoise the "index case" throughout the text to key in that the results being reported are for this case. The text would also benefit from calling out WHY the second population was sampled in the MM (Ex: "After the initial diagnosis of Adintovirus in the index case, a population of tortoises under human care were sampled." or something similar.
4. Abstract would benefit from a conclusion summary statement at end.
5. The text is still out of order and would benefit from re-ordering. For instance, the histopath results are reported before the surgical technique is described. Further, the methods for blood analysis should be in the MM, not the results.
6. Consider "under human care" as a replacement for "captive" whenever possible, as the zoo veterinary community is moving away from the latter term for welfare reasons.
7. My ethical/welfare concerns surround the surgical and analgesic treatment of this index case. Your report highlights that advancing reptile medicine involves utilization of more advanced molecular techniques, which is a stellar point. However, I counter, that to advance reptile medicine clinicians should also be pushing for a higher standard of care. This animal should have been intubated, as a surgical procedure with unknown bleeding potential was performed in its oral cavity. The fact that the tortoise maintained ventilation is great, however, not the only reason for intubation. Although analgesics were provided, they were minimal (NSAIDS) and very temporary (local block). Given that the tortoise did not eat for 5 days after the surgical procedure (line 207), I would postulate that there was pain and discomfort during the expected mucosal healing process. If this procedure were to have been performed in a "higher order" vertebrate (bird or mammal), these clinical decisions would be unacceptable. I push our entire clinical industry to drive reptile medicine forward. Although the manuscript is not necessarily about clinical medicine of individual animals, those who read the manuscript may emulate these behaviors. I recommend placing a note about the management of this case and how things could have been improved. This could be paired with how additional diagnostics (coelomic imaging/liver sampling) could add additional information in this case in a work up of systemic viral infection.
Comments on the Quality of English LanguageThere are still multiple areas where there are incomplete sentences (Ex: line 220-221), syntax errors (line 46), and overly wordy sentences that would benefit from revisions.
